# Elastocaloric, barocaloric and magnetocaloric effects in spin crossover polymer composite films

Klara Lünser [1], Eyüp Kavak [2,3], Kübra Gürpinar [3,4], Baris Emre [2], Orhan Atakol[4], Enric Stern-Taulats [1], Marcel Porta [5], Antoni Planes [1], Pol Lloveras [6], Josep-Lluís Tamarit [6] & Lluís Mañosa [1]✉

Giant barocaloric effects were recently reported for spin-crossover materials. The volume change in these materials suggests that the transition can be influenced by uniaxial stress, and give rise to giant elastocaloric properties. However, no measurements of the elastocaloric properties in these compounds have been reported so far. Here, we demonstrated the existence of elastocaloric effects associated with the spin-crossover transition. We dissolved particles of ([Fe(L)$_2$](BF$_4$)$_2$, [L=2,6di(pyrazol-1-yl)pyridine]) into a polymeric matrix. We showed that the application of tensile uniaxial stress to a composite film resulted in a significant elastocaloric effect. The elastocaloric effect in this compound required lower applied stress than for other prototype elastocaloric materials. Additionally, this phenomenon occurred for low values of strain, leading to coefficient of performance of the material being one order of magnitude larger than that of other elastocaloric materials. We believe that spin-crossover materials are a good alternative to be implemented in eco-friendly refrigerators based on elastocaloric effects.

Certain molecular complexes containing transition metal ions can undergo a spin-crossover (SCO) transition between low-spin (LS) and high-spin (HS) states[1]. At the SCO transition, these complexes experience significant changes in their magnetisation, volume and optical properties. The possibility of inducing the SCO transition by different external stimuli (temperature, light, pressure, etc.) gives rise to a variety of multifunctional properties with extensive value in multiple applications, such as memory, switching and sensing devices[2]. Currently, a vast amount of SCO complexes have been synthesised, most of which are powders or small single crystals. Many SCO transitions in these materials are first-order and involve a considerable latent heat. The use of the latent heat at the SCO transition has prompted thermal

applications of these compounds, such as thermal energy management[3] and giant barocaloric effects (BCEs)[4–6].

The caloric effect refers to the reversible thermal response of a given material to the application and removal of an external stimulus[7]. Depending on the nature of this stimulus, the different caloric effects are considered magnetocaloric, electrocaloric, barocaloric and elastocaloric for a magnetic field, for an electric field, for hydrostatic pressure and for uniaxial stress, respectively[8]. Although reports of several caloric effects date back more than one century[9,10], these effects received little attention from the scientific community until recently when it was shown that they can be used for near to room temperature refrigeration. Hence, solid-state cooling based on caloric

[1]Departament de Física de la Matèria Condensada, Facultat de Física, Universitat de Barcelona, Barcelona, Catalonia, Spain. [2]Department of Engineering Physics, Faculty of Engineering, Ankara University, Ankara, Turkey. [3]Graduate School of Natural and Applied Sciences, Ankara University, Ankara, Turkey. [4]Department of Chemistry, Faculty of Science, Ankara University, Ankara, Turkey. [5]Departament de Física Quàntica i Astrofísica, Facultat de Física, Universitat de Barcelona, Barcelona, Catalonia, Spain. [6]Grup de Caracterització de Materials, Departament de Física and Barcelona Research Center in Multiscale Science and Engineering, EEBE, Universitat Politècnica de Catalunya, Barcelona, Catalonia, Spain. ✉e-mail: lluis.manosa@fmc.ub.edu

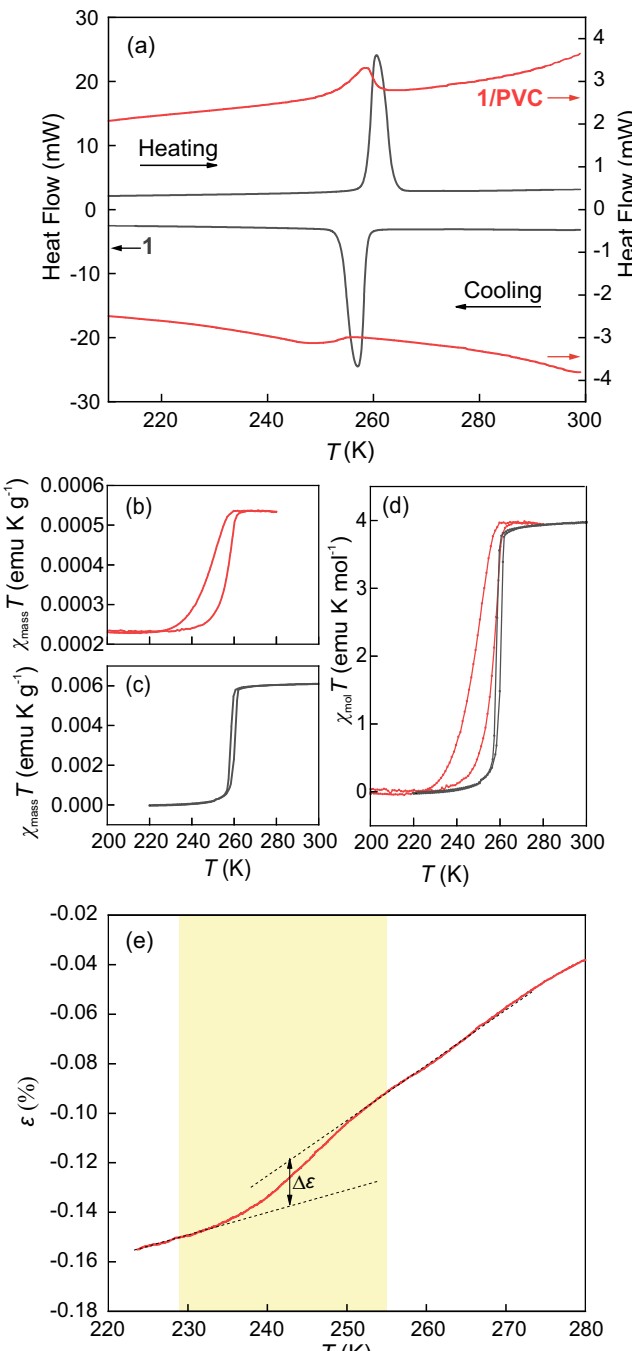

**Fig. 1 | Spin-crossover transition. a** Differential scanning calorimetry curves for pure **1** (black) and **1**/PVC (red) for heating (upper curves) and cooling (bottom curves) runs. Mass magnetic susceptibility data during cooling and heating of **1**/PVC (**b**, red curves) and pure **1** (**c**, black curves). **d** Molar magnetic susceptibility data (for **1**/PVC they have been corrected from the magnetic background arising from the non-transforming material; see the text for details). **e** Strain as a function of temperature for **1**/PVC. The coloured region corresponds to the region where the HS-to-LS transition occurs.

materials is a good alternative to replace current cooling systems that use harmful fluids (hydrofluorocarbons (HFCs))[11].

Caloric effects are intrinsic to most materials, but typically the thermal changes resulting from the application and removal of external fields are very weak and of limited technological interest. However, when the material is near a phase transition, these caloric effects are magnified. First-order phase transitions in which the latent heat

contributes greatly to the caloric effect are of interest, which gives rise to the giant and colossal caloric effects[7,8,12].

Among the several available caloric materials, barocaloric and elastocaloric materials have recently gained additional interest because of the much larger isothermal entropy changes achieved in this kind of material compared to their magnetocaloric and electrocaloric counterparts[13,14]. Some barocaloric materials exhibit colossal entropy changes with values above $100\,\mathrm{J\,kg^{-1}\,K^{-1}}$, which approach those of conventional HFCs[5,15,16]; however, implementing these materials in useful devices remains challenging. Compared to those in barocaloric materials, the entropy changes in elastocaloric materials are lower, but the materials are more compact, and uniaxial loads are more easily applied than hydrostatic pressure. To date, several heating and cooling prototypes have been reported based on the elastocaloric effect (eCE)[17–19]. It is acknowledged that any giant barocaloric material can exhibit giant eCEs when subjected to a uniaxial load[20]. Therefore, the possibility of underpinning the large entropy change involved in the phase transition of barocaloric materials through uniaxial stress appears to be a very attractive alternative to envisage compact and high-performing cooling devices. However, the powder nature of most giant and colossal barocaloric materials makes the application of uniaxial loads very challenging. Here we provide experimental evidence of the existence of giant eCEs associated with the SCO transition in samples subjected to uniaxial tensile stress. Therefore, we produced films of a polymeric matrix (polyvinyl chloride (PVC)) containing crystallites of a prototype barocaloric SCO compound ([Fe(L)$_2$](BF$_4$)$_2$, [L = 2,6di(pyrazol-1-yl)pyridine]). We show that the excellent barocaloric properties of SCO compounds are preserved in the SCO/PVC composite. By studying films subjected to uniaxial tensile stresses, we demonstrate remarkable eCEs associated with the SCO transition. The studied compound shows potential for future integration into cooling devices, providing research opportunities as the inert mass of the polymer currently impacts the cooling capacity. However, the specific elastocaloric values of the SCO are remarkable, with an isothermal entropy change of $\Delta S = 3.1\,\mathrm{J\,kg^{-1}\,K^{-1}}$ for stresses as low as $\sigma = 8\,\mathrm{MPa}$ and very low strain values $\varepsilon = 0.3\%$. To our knowledge, this is the first report of an eCE associated with the SCO transition. The elastocaloric performance characteristics of SCO compounds are more favourable than those of prototypical elastocaloric materials, such as shape memory alloys and elastomers. The required stresses are one order of magnitude lower than those required for shape memory alloys, and strains are several orders of magnitude lower than those of elastomers. Therefore, we anticipate that SCO compounds are excellent candidates for elastocaloric-based cooling technologies, especially in applications requiring low applied stresses. Notably, the techniques and results described in this work are not exclusive to SCO compounds but are expected to be extensive to a multitude of giant and colossal barocaloric materials.

## Results
The [Fe(L)$_2$](BF$_4$)$_2$, [L = 2,6di(pyrazol-1-yl)pyridine] compound (**1**) (Fig. S1) was synthesised as described in the "Methods" section, and the produced sample was analysed through infrared (IR) spectroscopy and nuclear magnetic resonance (NMR). The results presented in Figs. S2 and S3 corroborate the high quality of **1**. The compound was mixed with PVC, as described in the "Methods" section, to produce composite **1**/PVC films.

Atmospheric pressure calorimetric curves (heat flow vs. temperature) measured during cooling and heating are shown in Fig. 1a for pure **1** (black curves) and **1**/PVC with a 30% mass percentage of **1** (red curves). For comparison, differential scanning calorimetry (DSC) curves for pure PVC are presented in the Supplementary Material (Fig. S4a), which show the good stability of the polymer within the studied temperature range. For **1**/PVC, clear endothermic (heating) and exothermic (cooling) peaks are visible, corresponding to the LS-to-HS and HS-to-LS transitions of the SCO compound, respectively. The

presence of calorimetric peaks for **1**/PVC confirms that the SCO transition is preserved in the composite form. A perfect reproducibility of the transition is found, as evidenced by the DSC curves measured over ten consecutive cycles shown in Fig. S5a. The SCO transition in **1**/PVC is significantly different from that observed in the pure compound. The height of the calorimetric peaks is significantly reduced, the transition shifts towards reduced temperatures, the peak spreads over a broadened temperature domain, and an enlarged thermal hysteresis occurs. These differences are mostly attributed to the elastic interaction between **1** crystallites and the PVC matrix[21]. An appropriate model accounting for these interactions will be discussed in the following section. By suitable baseline correction and integration of the calorimetric curves, we obtained the transition entropy change ($\Delta S_t$) of the SCO transition (computed as the average between cooling and heating values). For **1**, $\Delta S_t = 92\,\mathrm{J\,kg^{-1}\,K^{-1}}$ which is in agreement with the data reported for the same compound in the literature[22]. For **1**/PVC, $\Delta S_t = 4.7\,\mathrm{J\,kg^{-1}\,K^{-1}}$. The value for the composite is relatively low due to the reduced mass of **1** in the sample. Considering that **1** accounts for 30% of the total mass, a value of $\Delta S_t = 27.6\,\mathrm{J\,kg^{-1}\,K^{-1}}$ is expected for the composite. However, the value computed from the recorded calorimetric curves ($\Delta S_t = 4.7\,\mathrm{J\,kg^{-1}\,K^{-1}}$) is significantly reduced, which indicates that not all **1** present in the composite undergoes the SCO transition; instead, a fraction of **1** remains in the HS state even at low temperatures. Considering the extensivity of the entropy, it is possible to evaluate the amount of **1** undergoing the SCO transition, by computing the ratio of $\Delta S_t$ obtained for **1**/PVC and for **1**. This ratio ($\frac{\Delta S_t^{1/PVC}}{\Delta S_t^1} = 0.051$) shows that only 5% of the total mass of the composite undergoes the SCO transition (which corresponds to $\sim\frac{1}{6}$ of the **1** mass in the composite). The occurrence of residual HS states at temperatures far below the SCO transition temperature is common in SCO complexes embedded in elastic matrices, and this phenomenon is attributed to surface effects[23,24].

To confirm the presence of HS **1** in the composite at low temperatures, we performed temperature-dependent X-ray measurements. The results are shown in the Supplementary Material in Figs. S6 and S7 for pure **1** and **1**/PVC, respectively. At room temperature, **1** exhibits a monoclinic structure (space group P2₁) with lattice parameters $a = 0.8504\,\mathrm{nm}$, $b = 0.8518\,\mathrm{nm}$, $c = 1.9078\,\mathrm{nm}$, and $\beta = 95.6°$ (determined from the Rietveld refinement of the data). Upon cooling, **1** undergoes an HS-to-LS transition. There is no change in the crystal symmetry, but there are changes in the lattice parameters for $a = 0.8523\,\mathrm{nm}$, $b = 0.8581\,\mathrm{nm}$, $c = 1.8571\,\mathrm{nm}$, and $\beta = 98°$ at $T = 255\,\mathrm{K}$, resulting in a $\frac{\Delta v}{v} = 2.3\%$ relative volume change at the SCO transition. The present results are in agreement with previously published data[4]. The room temperature X-ray pattern for **1**/PVC is shown in Fig. S7a. Clear diffraction peaks from **1** are visible above the polymeric background. The presence of these diffraction peaks confirms that **1** is in the form of crystallites dispersed within the PVC matrix. The presence of these crystallites is confirmed by scanning electron microscopy (SEM) imaging (Fig. S8). The diffraction peaks of **1** in the composite are broader than those corresponding to the pure compound, indicating that **1** crystallites are subjected to stress in the composite material. The diffraction pattern of **1**/PVC at a low temperature is shown in Fig. S7b. The data confirm that most of the compound remains in the HS state but very small peaks arising from a small fraction of **1** that has transformed to the LS state are visible (inset in Fig. S7b).

Isofield magnetisation measurements as a function of temperature are performed on heating and cooling **1** and **1**/PVC at selected values of the applied magnetic field. For comparison, the data for pure PVC are presented in Fig. S4b, which shows the diamagnetism of the polymer over the whole temperature range of interest. The magnetisation results for **1**/PVC are shown in the Supplementary Material (Fig. S9). From these data, we computed the temperature dependence of the magnetic susceptibility. For the different fields, the magnetic

susceptibility curves effectively collapse. The product of temperature and mass magnetic susceptibility as a function of temperature is shown in Fig. 1b for **1**/PVC and in Fig. 1c for **1**. In both cases, the data are computed by considering the total mass of the sample. The phase transition from HS to LS upon cooling and from LS to HS upon heating is clearly visible for both the **1** and **1**/PVC samples. For **1**/PVC, there is a significant magnetic response at low temperatures due to the fraction of **1** remaining in the HS state. The molar susceptibility data are shown in Fig. 1d. In the HS state, $\chi_{\mathrm{mol}}T = 3.9\,\mathrm{emu\,K\,mol^{-1}}$, which is in agreement with published data[22] for Fe$^{II}$ compounds with $S = 2$. For **1**/PVC, the molar susceptibility is computed by considering the mass of **1** that undergoes the SCO transition, which is derived from the calorimetric data (i.e. 5% of the total mass of the sample). For a good comparison with the pure compound, we subtract the magnetic background arising from the non-transforming compound **1**. Interestingly, the change in $\chi_{\mathrm{mol}}T$ at the SCO transition is $\sim 3.9\,\mathrm{emu\,mol^{-1}}$, which is in agreement with the behaviour expected for **1** transforming from the HS ($S = 2$) to the LS ($S = 0$) state. This result provides additional confirmation of the computational accuracy of the amount of **1** that undergoes SCO transition. The SCO transition in **1**/PVC is broader in temperature, which is in alignment with the calorimetric data; overall, **1**/PVC shifts to relatively low temperatures and has a relatively large thermal hysteresis.

The length changes of a **1**/PVC composite film are measured as a function of temperature. Figure 1e shows the temperature dependence of the strain $\varepsilon = \frac{l - l_0}{l_0}$ (where $l_0$ is the total length of the film at room temperature) while cooling the sample through a temperature range encompassing the SCO transition. Additional measurements for different values of the applied force are shown in the Supplementary Materials (Fig. S5b). A noticeable behaviour change is observed over a temperature range that coincides with the extension of the calorimetric peak (indicated by the coloured area in Fig. 1e), which is likely due to the volume change of **1** undergoing the HS-to-LS transition. The strain at high temperatures is greater than that at low temperatures, which is consistent with the relatively large volume of the HS phase in **1**. The estimated strain change $\Delta\varepsilon \sim 0.04\%$ is consistent with the relative volume change of $\frac{\Delta v}{v} = 2.3\%$ when considering the conditions of isotropy $\Delta\varepsilon = \frac{1}{3}\frac{\Delta v}{v}$ and that 5% of the total mass is the transforming material.

Isothermal magnetisation measurements as a function of magnetic field for **1**/PVC are shown in Fig. 2a, b for increasing and decreasing temperature, respectively. For a few selected temperatures, we verify that the data recorded from increasing fields coincide with those recorded from decreasing fields. Experiments are conducted for increasing magnetic fields at all temperatures. The data measured for **1** are shown in the Supplementary Material (Fig. S10). From the magnetisation data, we compute the isothermal entropy change ($\Delta S_{\mathrm{MCE}}$) corresponding to the magnetocaloric effect (MCE) for **1**/PVC and pure **1** as follows:

$$\Delta S_{MCE}(T, 0 \to \mu_0 H) = \mu_0 \int_0^H \left(\frac{\partial M}{\partial T}\right)_H dH \qquad (1)$$

The results are shown in Figs. 2c and S11. The MCE associated with the SCO transition is preserved in SCO/polymer composites. Figure 2c shows the MCE data for a 7 T field. The red symbols and lines correspond to **1**/PVC, while the black symbols and lines correspond to **1**. The data for **1**/PVC are computed by considering the mass of **1** that undergoes the SCO transition. In all cases, the open symbols correspond to cooling (HS-to-LS transition) while the solid symbols correspond to heating (LS-to-HS transition).

The MCE is inverse (i.e. the entropy increases with increasing magnetic field), in accordance with the relatively large magnetisation of the high-temperature phase. As expected from the weak magnetic behaviours of SCO compounds, $\Delta S_{\mathrm{MCE}}$ values for **1** and **1**/PVC are small. The MCE in **1** achieves higher entropy values and extends over a narrower temperature domain than that in **1**/PVC, which extends towards

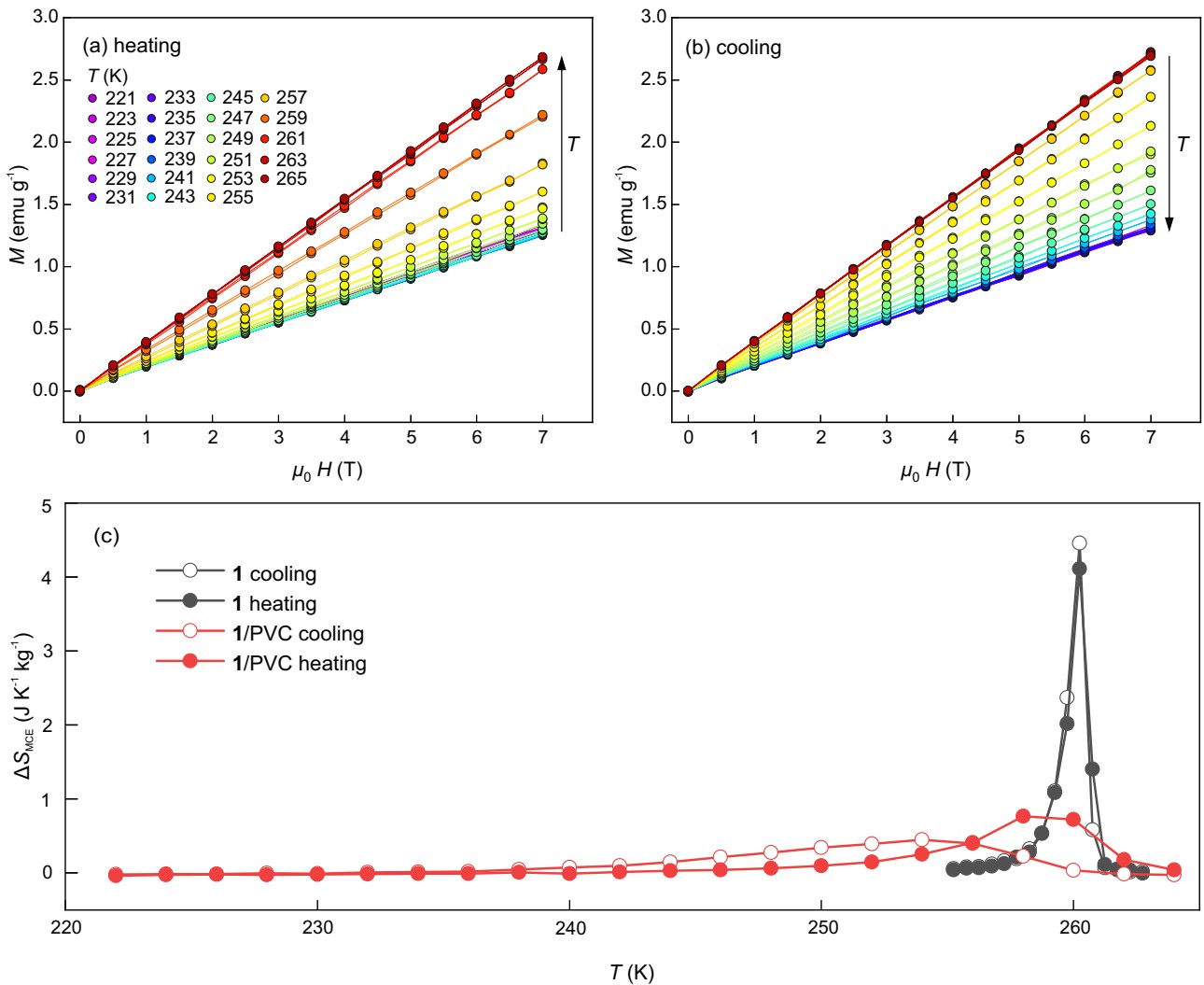

**Fig. 2 | Magnetocaloric effect.** Isothermal magnetisation as a function of magnetic field for selected increasing (**a**) and decreasing (**b**) temperatures for **1**/PVC. The same colour code applies to the temperature values for the two panels. **c** Isothermal entropy change for the application of a 7 T magnetic field for **1** (black symbols and lines) and **1**/PVC (red symbols and lines). The open symbols correspond to decreasing temperatures, while the solid symbols correspond to increasing temperatures. Lines are guides to the eye.

the low-temperature region. These differences are consequences of the extended SCO transition in **1**/PVC resulting from the elastic interactions between the **1** crystallites and the polymer matrix. Interestingly, the refrigerant capacity (RC) computed as the area below the $\Delta S_{MCE}$ vs. $T$ curve is similar for **1** and **1**/PVC (RC ~ 5.2 J kg$^{-1}$).

While MC effects in SCO compounds are weak, these materials exhibit giant BCEs. To determine whether these giant BCEs are preserved when SCO compounds are dispersed in a polymeric matrix, we performed calorimetric measurements at selected hydrostatic pressures in **1**/PVC. The small percentage of material that undergoes the SCO transition for the **1**/PVC and the weak sensitivity of the bespoke calorimeter under pressure results in a weak calorimetric signal at the HS−LS transition. Nevertheless, endothermal and exothermal peaks corresponding to the LS-to-HS and HS-to-LS transitions, respectively, can be identified, and the baseline corrected curves are shown in Fig. S12 (where we have only considered the mass of **1** undergoing the SCO transition). Integration of these curves renders the transition entropy change. Within our experimental errors, we do not detect any dependence of $\Delta S_t$ on pressure.

A combination of these calorimetry data with specific heat ($C_p$) measurements enables the determination of the entropy as a function of temperature and pressure (the entropy curves $S(T, p)$ are referenced

to a value at a given temperature $T_0$ and to atmospheric pressure). Furthermore, for compressible materials, the BCE beyond the phase transition must be considered, and it can be computed from the thermal expansion $\beta_v$ of the high- and low-temperature phases. Details of all these calculations are provided in the Supplementary Material. For **1**/PVC, we use $C_p$ (Fig. S13) and $\beta_v$ for the pure compound (with $\beta_v^{HS} = 2.0 \times 10^{-4}$ K$^{-1}$ and $\beta_v^{LS} = 2.8 \times 10^{-4}$ K$^{-1}$ [4]). The obtained $S(p, T)$ curves are shown in Figs. 3a (cooling runs) and 3b (heating runs). The pressure-induced isothermal entropy change corresponding to the pressure application is readily obtained from these curves as follows:

$$\Delta S_{BCE}(T, 0 \rightarrow p) = S(T, p) - S(T, 0) \qquad (2)$$

(equivalent expressions are true for pressure removal).

The results for $\Delta S_{BCE}$ are shown in Fig. 3c, d. The BCE is conventional (i.e. the entropy decreases with the application of pressure) in accordance with the reduced volume of the low-temperature phase. Figure 3c shows the $\Delta S_{BCE}$ resulting from the application of pressure, which promotes the HS-to-LS transition. Figure 3d shows the $\Delta S_{BCE}$ resulting from the removal of pressure, which promotes the LS-to-HS transition.

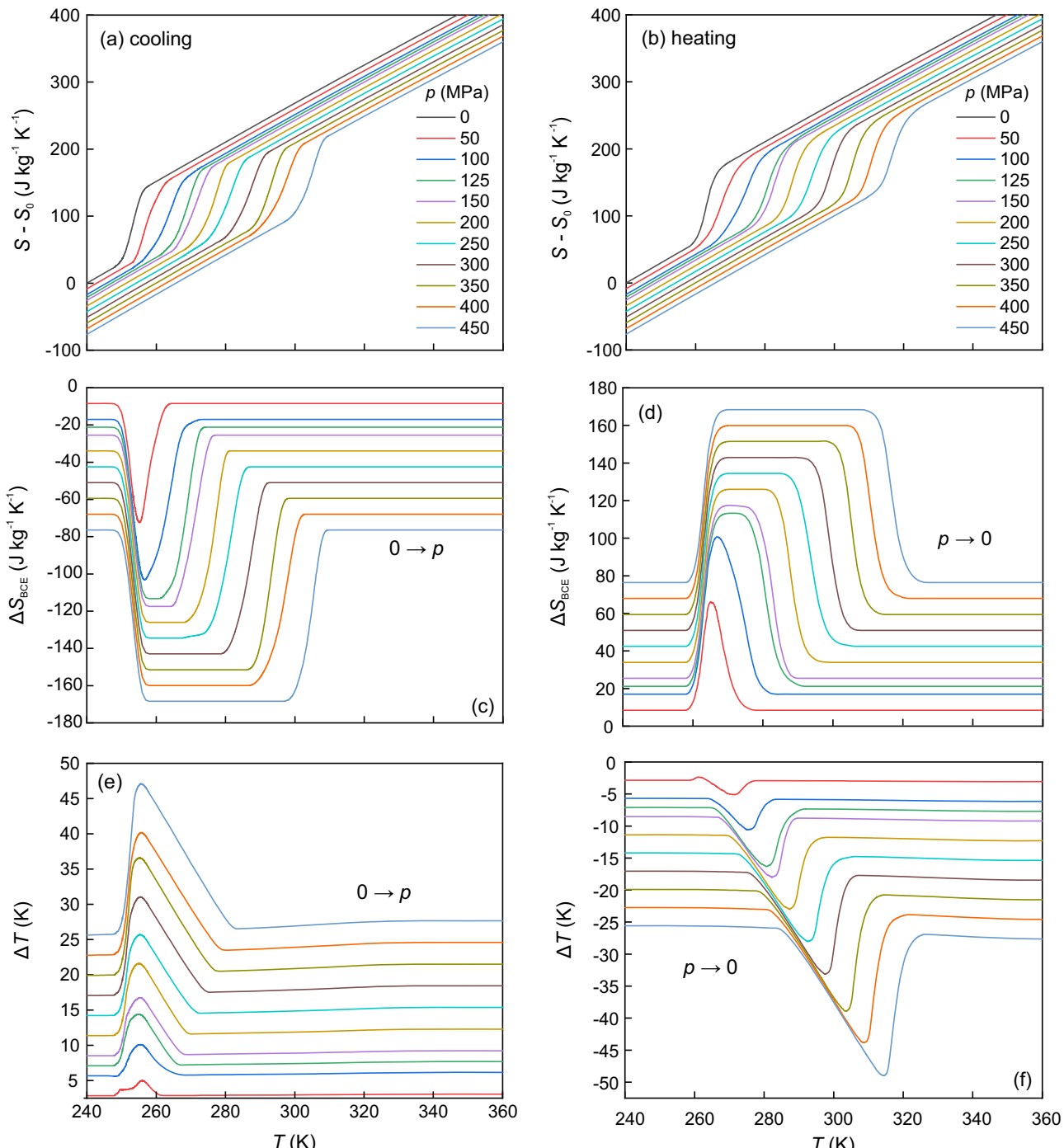

**Fig. 3 | Barocaloric effect.** Temperature-dependent isobaric entropy curves at selected pressure values, upon cooling (**a**) and heating (**b**) **1**/PVC. Barocaloric iso-thermal entropy changes for compression (**c**) and decompression (**d**) and adiabatic temperature changes for compression (**e**) and decompression (**f**) as a function of temperature at selected pressure values. The same colour code for pressure applies to all panels.

By inverting the $S(T, p)$ curves, it is also possible to compute the pressure-induced adiabatic temperature change as follows:

$$\Delta T_{BCE}(S, 0 \rightarrow p) = T(S, p) - T(S, 0) \qquad (3)$$

where the data are typically plotted as a function of the temperature before the application of pressure. Equivalent expressions are true for pressure removal.

The results for **1**/PVC are shown in Fig. 3e, f for the application and removal of pressure, respectively corresponding to the HS-to-LS and LS-to-HS transitions.

The $\Delta S_{BCE}$ and $\Delta T_{BCE}$ values for **1**/PVC are very large ($|\Delta S_{BCE}| = 168\,\mathrm{J\,kg^{-1}\,K^{-1}}$ and $|\Delta T_{BCE}| = 49\,\mathrm{K}$ for a 450 MPa pressure change) and range among the highest values for any caloric material[14]. Overall, the present data for **1**/PVC are comparable to the reported barocaloric data for the pure **1** compound[4], although in our study a greater pressure range has been explored. $d(\Delta T_{ad})/dp = 100\,\mathrm{K\,GPa^{-1}}$ is reported in the literature[4], which is in very good agreement with the $\Delta T = 10\,\mathrm{K}$ value for **1**/PVC for an applied pressure of 100 MPa. Notably, the barocaloric properties of **1** are preserved when dispersed in PVC. However, the values provided refer to the actual mass undergoing the SCO transition. These values decrease

since a portion of **1** does not undergo the SCO transition, and the polymer adds an inert mass.

While MCE and BCE are reported for pure SCO compounds, the powder nature of these materials prevents the application of uniaxial stresses and no studies on the possible eCE of SCO have been undertaken. Dispersing an SCO compound in a polymeric matrix enables the application of uniaxial stresses and the identification of an eCE associated with the SCO transition, which, to the best of our knowledge, has not yet been reported. Therefore, we perform isothermal stress vs. strain measurements on a **1**/PVC film over a temperature range covering the HS-to-LS transition of **1**. The experiments are restricted to low applied loads to prevent damage to the samples. The stress ($\sigma$) is computed as the ratio between the applied force and the cross-sectional area of the sample (which is assumed to be constant).

The stress dependence of the strain at selected temperatures is shown in Fig. 4a for **1**/PVC. The data are obtained by increasing and decreasing stress, and the resulting hysteresis increases with increasing temperature but remains low for the studied temperature range. At high temperatures, PVC approaches the glass transition and melting point with increasing irreversibility, as evidenced by the increasing hysteresis between the loading and unloading runs.

The stress-induced entropy change ($\Delta S_{eCE}$) can be computed from stress–strain curves as follows:

$$\Delta S_{eCE}(T, 0 \rightarrow \sigma) = v \int_0^\sigma \left(\frac{\partial \varepsilon}{\partial T}\right)_\sigma d\sigma \qquad (4)$$

where $v$ is the specific volume ($v = 0.625\ \mathrm{cm^3\ g^{-1}}$ for the HS state).

The results for selected values of the applied stress are shown in Fig. 4b (for comparison, the data for pure PVC are shown in Fig. S4c). The marked peak of $\Delta S_{eCE}$ over the temperature region where the LS-to-HS transition of **1** occurs represents the first experimental evidence of an eCE associated with an SCO transition. The eCE is inverse (i.e. the entropy increases with increasing applied stress). This phenomenon is in agreement with the large strain of the HS phase (Fig. 1e). The tensile uniaxial stress stabilises the large strain phase. The maximum values for $\Delta S_{eEC}$ increase with increasing stress, with $\Delta S_{eCE} = 3.1\ \mathrm{J\ kg^{-1}\ K^{-1}}$ for $\sigma = 8\ \mathrm{MPa}$. Notably, despite the low value of the applied stress, the elastocaloric entropy change is relatively large, which proves that SCO compounds are prone to exhibit giant eCEs.

## Discussion

To explain the shift of the phase transition to relatively low temperature when the SCO compound is embedded into PVC, we propose a Ginzburg–Landau model of an inclusion exhibiting a volumetric phase transition embedded into an elastic matrix. The model simulation reveals that the elastic energy of the matrix after the phase transformation of the inclusion scales linearly with the volume of the inclusion (Supplementary Materials, Fig. S14). The free energy of the inclusion, $F_{inc}$, and the deformed matrix, $F_{mat}$, can be approximated as follows:

$$F_{inc} \approx \left[\frac{1}{2}A(T - T_c)e_1^2 + \frac{1}{3}\zeta e_1^3 + \frac{1}{4}\gamma e_1^4\right]\Omega$$
$$F_{mat} \approx Be_1^2\Omega, \qquad (5)$$

where $A$, $\zeta$ and $\gamma$ are related to the second and higher order elastic moduli of the SCO material, $B$ is a parameter that depends on the bulk modulus of the matrix and on the shape of the inclusion. $e_1$ is the transformation strain within the inclusion, which is assumed to be homogeneous, and $\Omega$ is the volume of the inclusion. Thus, the total free energy can be written as follows:

$$F = F_{inc} + F_{mat} \approx \left\{\frac{1}{2}A[T - (T_c - 2B/A)]e_1^2 + \frac{1}{3}\zeta e_1^3 + \frac{1}{4}\gamma e_1^4\right\}\Omega. \qquad (6)$$

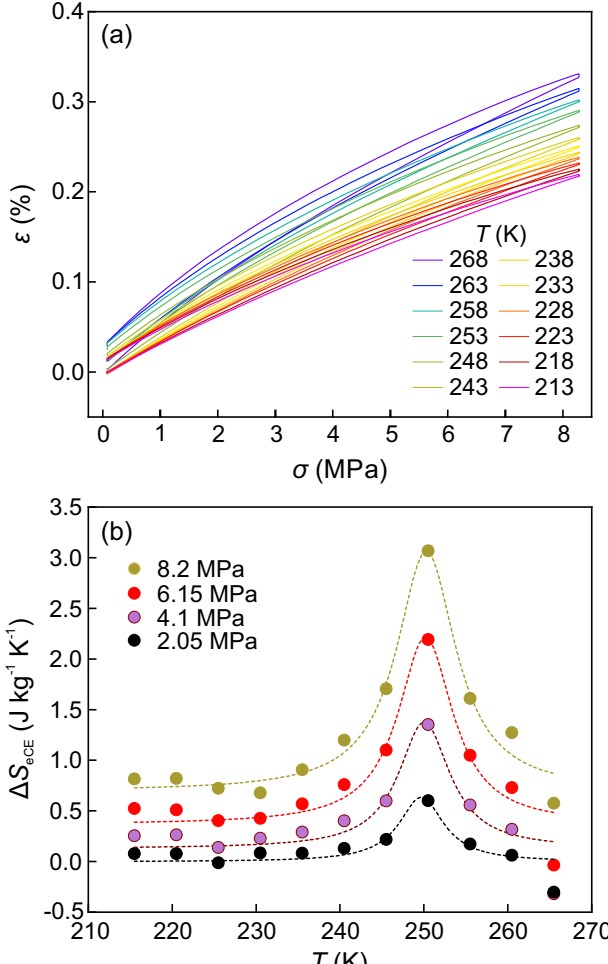

**Fig. 4 | Elastocaloric effect. a** Isothermal strain as a function of applied uniaxial tensile stress at selected temperatures. **b** Stress-induced elastocaloric isothermal entropy change for selected values of applied uniaxial tensile stress. Dashed lines are visual guides.

The result is that the elastic energy of the matrix shifts the stability limit of the high-temperature phase of the inclusion, $T_c$, to relatively low temperature by $2B/A$. Consequently, the phase transformation of the inclusion also shifts to a relatively low temperature. In addition, if the value of parameter $B$ is slightly different for each inclusion (due to fluctuations in the stiffness of the matrix or due to the shape of the inclusion), then the transition temperature is different for each inclusion, which smooths the macroscopic phase transformation.

This result can be used to estimate the fraction of the system in the HS phase during cooling and heating cycles, which should qualitatively reflect the magnetic susceptibility obtained in experiments. Therefore, we consider an ensemble of isolated inclusions, each with a slightly different value of $T_c$, to avoid sharp macroscopic transitions. During the cooling process, only those inclusions with a value of $T_c' = T_c - 2B/A$ below the current temperature are assumed to remain in the HS phase. In contrast, during the heating process, only inclusions with a stability limit of the LS phase, $T_{LS}' = T_c' + \frac{1}{4}\zeta^2/A\gamma$, below the current temperature are assumed to have transformed into the HS phase. The curves obtained are shown in Fig. 5a for SCO embedded into the polymer and in Fig. 5b for pure SCO. The probability densities of the $T_c$ and $B$ values are assumed to be Gaussian, and the parameters of the model are refitted to the experimental data. Differentiating these curves yields a behaviour qualitatively similar to the heat flow curves obtained in

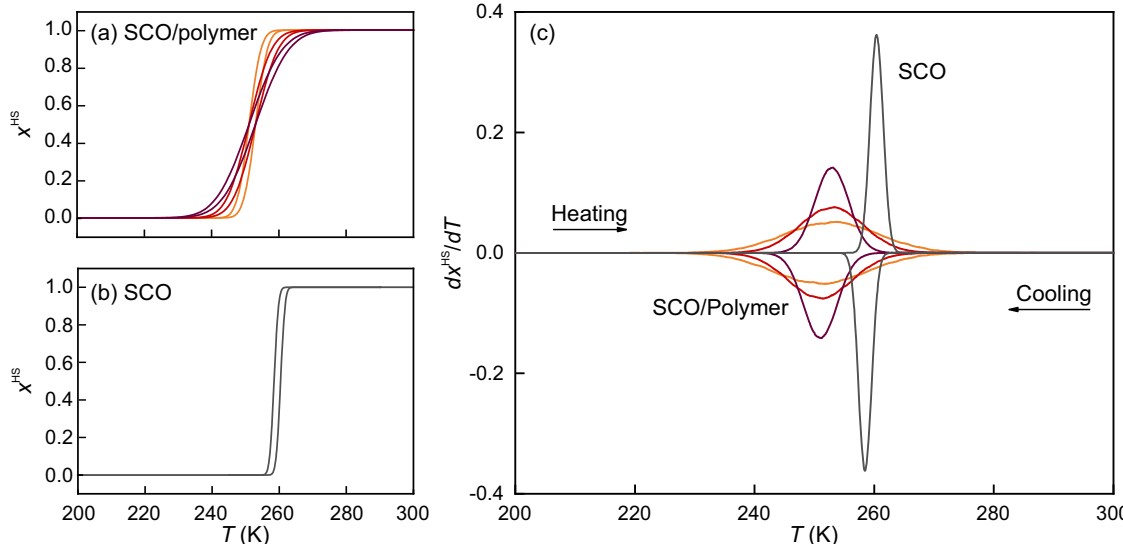

**Fig. 5 | Model results.** Fraction of the high-temperature phase vs. temperature during cooling and heating for inclusions embedded in an elastic matrix (**a**) and for pure material (**b**). The results for the inclusions are given for different values of the variance in parameter $B$. **c** Temperature derivative of the fraction of the high-temperature phase vs. temperature during cooling and heating for a pure material and for inclusions embedded in an elastic matrix. The results for the inclusions are given for different values of the variance in parameter $B$.

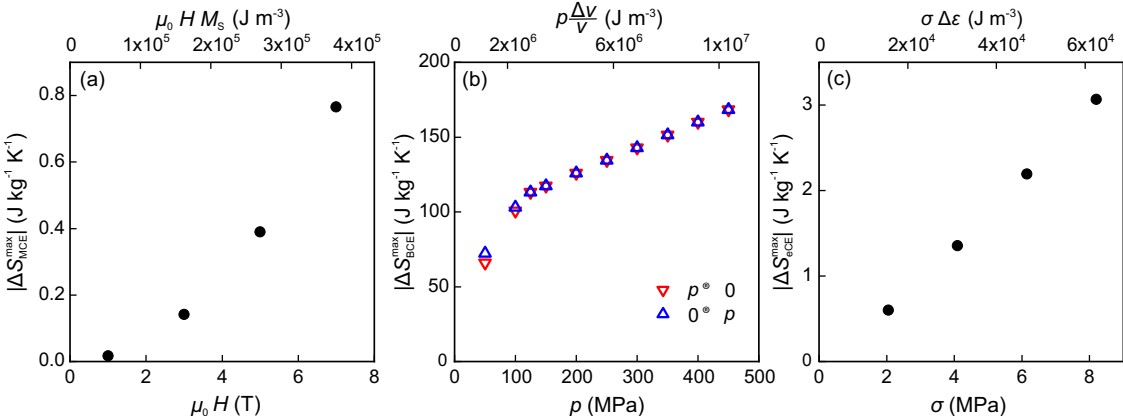

**Fig. 6 | Caloric isothermal entropy changes.** Maximum absolute value of the isothermal entropy change for magnetocaloric (**a**), barocaloric (**b**) and elastocaloric (**c**) effects as a function of magnetic field, hydrostatic pressure and uniaxial tensile stress, respectively (bottom axis), and as a function of a characteristic energy (upper axis, see text for details).

experiments, as shown in Fig. 5c. In this figure, the fact that the tail of the peak centred at $T_c'$ extends to temperatures above $T_c$ is not physically meaningful, and results from the long tails of the Gaussian distribution. This is a minor effect that does not affect the main results of the model.

It is expected that the value of the polymer bulk modulus significantly affects the eCE properties of the composite. For very stiff polymers (large $B$ values), the elastic energy required to deform the matrix is large. If the free energy difference between the HS and LS phases is not large enough to overcome the increase in (elastic) energy, the SCO transformation will be arrested and no eCE occurs. Conversely, very soft polymers (with low $B$ values) cannot transmit the external force to the SCO particles, and no eCE associated with the SCO occurs. Hence, $B$ must be within an appropriate range of values such that the polymer can transmit the external force, and it must not impede the transformation of SCO particles.

The results presented in the previous section demonstrate that the caloric properties of SCO compounds are preserved when they are dispersed into a polymeric matrix. For each caloric effect (MCE, BCE and eCE) the isothermal entropy change increases in absolute value

when the external stimuli (magnetic field, hydrostatic pressure and uniaxial stress) increase, as illustrated in Fig. 6. This figure shows the dependence of $|\Delta S_{\text{MCE}}^{\text{max}}|$, $|\Delta S_{\text{BCE}}^{\text{max}}|$ and $|\Delta S_{\text{eCE}}^{\text{max}}|$ on the magnetic field, hydrostatic pressure and uniaxial tensile stress, respectively. To allow a good comparison for the different caloric effects, data have been plotted in terms of a characteristic energy given by $\mu_0 H M_s$ for the MCE, $p\frac{\Delta v}{v}$ for the BCE and $\sigma \Delta \varepsilon$ for the eCE where $M_s$ corresponds to the saturation magnetisation (estimated assuming a Brillouin dependence for the magnetisation), $\frac{\Delta v}{v}$ is the relative volume change at the SCO transition and $\Delta \varepsilon$ is the strain change at the SCO transition. The MCE (Fig. 6a) shows a small value for the magnetocaloric strength, $\frac{|\Delta S_{\text{MCE}}^{\text{max}}|}{\Delta(\mu_0 H)} = 0.1 \, \text{J kg}^{-1} \, \text{K}^{-1} \, \text{T}^{-1}$, in accordance with the weak magnetism of SCO compounds. For the BCE two regimes are clearly visible: $|\Delta S_{\text{BCE}}^{\text{max}}|$ sharply increases at low pressures ($p \leq 100$ MPa), showing a moderate linear increase at high pressures. A sharp increase is associated with the SCO transition, resulting from the increase in the fraction of **1** undergoing the HS-to-LS transition as the pressure increases. Once the SCO is completed, $|\Delta S_{\text{BCE}}^{\text{max}}|$ increases due to the contribution from

## Table 1 | Elastocaloric properties of prototype elastocaloric materials

| Material | $\Delta S$ | $\Delta \sigma$ | $\Delta \varepsilon$ | $\frac{\Delta S}{\Delta \sigma}$ | $\frac{\Delta S}{\Delta \varepsilon}$ | Ref |
|---|---|---|---|---|---|---|
| | J kg$^{-1}$ K$^{-1}$ | MPa | % | J kg$^{-1}$ K$^{-1}$ MPa$^{-1}$ | J kg$^{-1}$ K$^{-1}$ | |
| Ni–Ti | 35 | 800 | 3 | 0.04 | 11 | [31] |
| Cu–Zn–Al | 21 | 120 | 8 | 0.02 | 2.6 | [32] |
| Natural rubber | 80 | 4 | 400 | 20 | 0.13 | [33] |
| **1**/PVC | 3.1 | 8 | 0.3 | 0.4 | 10 | This work |

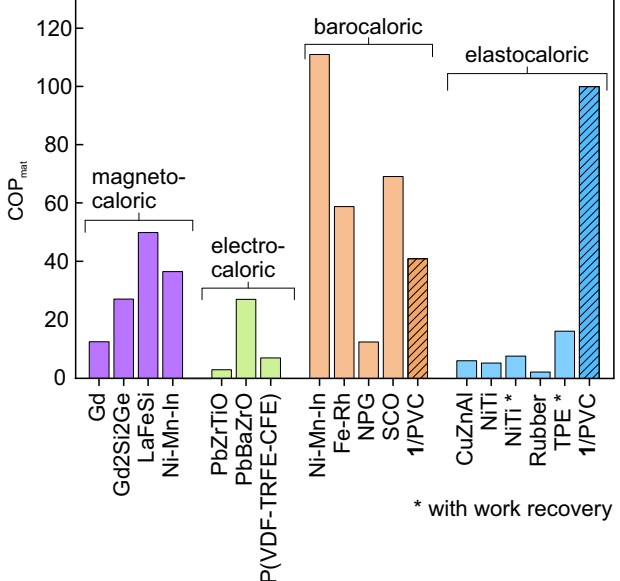

**Fig. 7 | Coefficient of performance.** Materials coefficient of performance for prototype magnetocaloric (purple), electrocaloric (green), barocaloric (orange) and elastocaloric (cyan) materials. The data for **1**/PVC correspond to the stripped bars. The asterisk indicates those cases where work recovery has been assumed in the computation of the coefficient. Data from refs. 5,16,25,26,33.

thermal expansion. The barocaloric strength $\frac{|\Delta S_{BCE}^{max}|}{\Delta p} = 1$ J kg$^{-1}$ K$^{-1}$ MPa$^{-1}$ is large, and is comparable to that of the best (colossal) barocaloric materials, such as plastic crystals ($\frac{|\Delta S_{BCE}^{max}|}{\Delta p} = 1.78$ J kg$^{-1}$ K$^{-1}$ MPa$^{-1}$)[16].

With regard to the eCE, the linear increase in $|\Delta S_{eCE}^{max}|$ with respect to $\sigma$ gives rise to a large elastocaloric strength of $\frac{|\Delta S_{eCE}^{max}|}{\Delta \sigma} = 0.4$ J kg$^{-1}$ K$^{-1}$ MPa$^{-1}$. Interestingly, the elastocaloric strength approaches one third of the barocaloric strength, which is the value that is expected by assuming an isotropic distribution of SCO crystallites and by considering that the structural change at the SCO transition is described by a volume change in the unit cell. In the present work, we have restricted our study to low-stress values, but the large elastocaloric strength anticipates giant eCEs for SCO/polymer composites which might support larger stresses. In Table 1, we compare the reported elastocaloric properties of **1**/PVC to those of prototype elastocaloric materials such as shape memory alloys and natural rubber. On the one hand, shape memory alloys require large stresses and the elastocaloric strength in terms of stress is quite low. However, the moderate deformation of these compounds results in favourable elastocaloric strength in terms of strain. On the other hand, the eCE in rubber occurs at low stress with a high elastocaloric strength. The large strain involved in the eCE results in a very low elastocaloric strength in terms of strain. Our **1**/PVC film shows a significant elastocaloric response at low values of stress and strain, exhibiting a good elastocaloric strength in terms of the two quantities. Interestingly, a combination of low applied stress and small strain change provides **1**/PVC

with a remarkably large materials coefficient of performance (COP$_{mat}$), which is defined as the ratio between the exchanged heat and the work needed to achieve the caloric effect (this ratio is also known as the material efficiency $\eta$[25–27]). In Fig. 7, we show a comparison of the COP$_{mat}$ values for selected prototype magnetocaloric, electrocaloric, barocaloric and elastocaloric materials. Several data have been taken from previous reviews[25,26]. We update the plot with values for recently reported giant and colossal barocaloric and elastocaloric materials, and present results for **1**/PVC. It has been argued that elastocaloric materials exhibit very poor COP$_{mat}$[25]. However, for **1**/PVC, COP$_{mat}$ = 100, which is a value that places SCO compounds among the best-performing caloric materials in terms of this parameter.

To sum up, we have studied the SCO transition and related caloric properties of a prototypical SCO compound [Fe(L)$_2$](BF$_4$)$_2$ and a PVC polymer composite film. The elastic interactions between the SCO crystallites and the polymeric matrix cause the transition to shift towards reduced temperatures and extend the temperature range of the SCO transition.

The caloric properties associated with the SCO transition are preserved in the composite film. The MCE in the film is weaker with lower values of the isothermal entropy change but spreads over a broader temperature window than that in the pure compound. The specific BCE exhibits isothermal entropy and adiabatic temperature changes comparable to those of the pure SCO compound. These effects are similar to those of the best caloric materials.

A key feature of SCO and polymer films is the possibility of applying uniaxial stresses to the SCO compound, which is challenging to achieve in pure SCO materials due to their powder or small single crystalline nature. We have shown that the application of very low uniaxial stresses to SCO/polymer films results in a remarkable eCE arising from the SCO transition. The eCE found in this study for SCO/polymer composite films shows a variety of advantages over prototypical elastocaloric materials, such as shape memory alloys and elastomers. In our material, a significant eCE can be observed for very low values of stress and strain. This effect differs from shape memory alloys, which require very high stresses, and from elastomers, which experience considerable strain. Notably, a combination of low stress and low strain in an SCO/polymer composite film results in a COP$_{mat}$ one order of magnitude larger than that of typical elastocaloric materials, with a value comparable to that of the best-performing caloric (barocaloric) materials.

In this work, we have shown the potential of SCO/polymer composite films as caloric materials for environmentally friendly solid-state refrigeration. It must be mentioned, however, that the studied material can be significantly optimised. The entropy and temperature changes reported in this study correspond to the SCO compound undergoing the SCO transition. For practical applications, these values are obviously significantly reduced in cooling devices because the quantity of non-transforming compounds and the inert mass of the polymer must be considered. The vast amount of SCO compounds reported thus far and the availability of many different polymers have extensive research prospects, with plenty of room for optimising the caloric performance of SCO/polymeric composites. A possible method for improvement is to identify the best preparation methods that increase the fraction of SCO compounds undergoing the SCO transition. Interesting alternatives for optimisation can be the use of polymers that support large stresses (resulting in relatively large elastocaloric entropy changes) and polymers with high thermal conductivities (resulting in accelerated heat exchange in refrigeration cycles). Moreover, the film geometries of the synthesised materials allow for the easy application of out-of-plane loads to drive caloric effects. This ease might be advantageous for applications as the film might obtain ideal heat transfer conditions during this mode of compression. Notably, the compact and flexible nature of SCO/polymer films opens up the possibility of other mechanocaloric effects associated with inhomogeneous stresses, such as the recently reported flexocaloric[28] and twistcaloric[29] effects.

In conclusion, SCO/polymer composites are a new family of giant mechanocaloric materials. The techniques and results presented for SCO compounds can be extended to a broad variety of giant and colossal BCE materials, such as plastic crystals. Our study opens new avenues of research that may lead to the discovery of materials with excellent caloric performance suitable for use in efficient and clean cooling devices.

## Methods

### Sample preparation

**Synthesis of [L][L = 2,6di(pyrazol-1-yl)pyridine].** The ligand (L) was synthesised in two steps according to the literature[30]. A total of 6.80 g (0.10 mol) of pyrazole was dissolved in diethylene glycol dimethyl ether (50 cm$^3$) in a two-necked round bottom flask at room temperature. To this solution, 2.30 g (0.10 mol) of metallic sodium was added to small pieces under an argon atmosphere, and the solution was stirred for ~3–4 h to obtain sodium pyrazolate. Next, 7.4 g (0.05 mol) of 2,6-dichloropyridine was added to this solution, which was heated under reflux conditions for 12 h and allowed to cool to room temperature. Distilled cold water (300 cm$^3$) was added, and the solid formed was filtered by suction and purified with a hot methanol–water mixture. The product was a bright white crystal. The yield was 72%, and the melting point ranged from 136 to 138 °C.

**Synthesis of the [Fe(L)$_2$](BF$_4$)$_2$ complex.** Fe(BF$_4$)$_2$·6H$_2$O (0.338 g, 0.001 mol) was dissolved in MeOH (25 ml). In another flask, the ligand (0.422 g, 0.002 mol) was dissolved in 40 ml of MeOH with heating, and the solutions were combined and stirred while heating for 5 min. The resulting crystalline yellow complex was filtered and air-dried. The yield was 72%. An elemental analysis was performed for [Fe(L)$_2$](BF$_4$)$_2$. The calculated elemental composition was as follows: C, 40.51; H, 2.78; N, 21.47; Fe, 8.56. The measured elemental composition was as follows: C, 39.82; H, 2.61; N, 21.07; and Fe, 8.59.

**Synthesis of the SCO/PVC film.** The polymer used was PVC purchased from Fluka Honeywell (Product No. 81392), corresponding to CAS No. 900-86-2, with a molecular weight of 62.49822 g mol$^{-1}$ and a density of 1.4 g ml$^{-1}$ (at 25 °C). The solution was obtained by dissolving the PVC polymer in tetrahydrofuran (THF), and the [Fe(L)$_2$](BF$_4$)$_2$ in methanol and mixing them together. The prepared solution was spread homogeneously in a petri dish using the drop-casting method. Subsequently, the sample was air-dried and further dried in an oven at 50 °C for a long time while mass loss was evaluated. The absence of THF was confirmed by thermogravimetric analysis between ambient conditions and 120 °C. The largest amount of SCO compound we successfully dispersed into the PVC matrix was 30%.

### Sample characterisation

IR spectra were recorded using ATR equipment on a Shimadzu infinity model FTIR device between 600 and 4000 cm$^{-1}$, and NMR spectra were obtained using a Bruker Advance 500 NMR spectrometer in d$_6$-DMSO.

X-ray diffraction experiments were performed on **1** powder and **1**/PVC films at various temperatures and at atmospheric pressure using a PANalaytics X'pert PRO MPD system with a PIXcel detector with an active length of 3.347° in transmission geometry. The powder of **1** was placed in a spinner glass capillary sample holder (Hilgenberg glass no. 14 capillary with a diameter of 1.0 mm), and a 20 × 2 × 0.2 mm$^3$ piece of **1**/PVC film was used directly as the capillary.

For SEM experiments, a piece of **1**/PVC film was stuck onto carbon tape and then coated with carbon. The image was taken with an acceleration voltage of 5 kV using secondary electrons.

### Calorimetric measurements

Atmospheric pressure DSC experiments were performed using a TA Instruments Q2000 calorimeter at a scanning rate of 10 K min$^{-1}$ on 19.7 mg of pure **1** compound, 15.7 mg of **1**/PVC film and 8.76 mg of pure PVC. Heat capacity ($C_p$) measurements were performed via modulated calorimetry using a DSC Q2000 from TA Instruments, with a modulation amplitude of ±0.5 K, a period of 60 s and a heating rate of 1 K min$^{-1}$. High-pressure differential thermal analysis was conducted using a customised calorimeter consisting of a Cu-Be MVI-30 high-pressure cell (Unipress, Poland) operating between 205 and 393 K and at pressures reaching 600 MPa. Several sheets of **1**/PVC (~ 100 mg overall mass) were put in direct close contact with Peltier modules acting as calorimetric sensors by using Teflon tape. The temperature was recorded by a type-T thermocouple attached to the sample. The pressure-transmitting fluid used was DW-Therm M90.200.02 (Huber). The temperature was controlled using an external thermal bath (Lauda Proline RP 1290) connected to a jacket surrounding the high-pressure cell, and quasi-isobaric runs were conducted at typical scanning rates of ~ 1–3 K min$^{-1}$.

### Magnetic measurements

Magnetic measurements were performed with a superconducting quantum interference device magnetic property measurement system (MPMS). For pure **1**, ~ 8.58 mg was packed in a polycarbonate capsule and placed in a vertical plastic straw. For **1**/PVC, six film pieces with dimensions of 6.6 × 2.8 × 0.2 mm$^3$ were stacked and tightly and vertically attached to a vertical plastic straw through silk wires. For pure PVC, a 22.93 mg sample was placed in a vertical plastic straw. The plastic straw was placed in the MPMS sample chamber. Isofield magnetisation measurements at selected applied magnetic fields were conducted in the temperature range of 200–300 K at a scanning rate of 2 K min$^{-1}$. Isothermal magnetisation measurements were performed in the range of 0–7 T at 0.5 T steps in the range of 220–265 K upon increasing (heating runs) and decreasing (cooling runs) the temperature at 2 K intervals.

### Mechanical measurements

Mechanical measurements were performed using a dynamic mechanical analyser Q800 from TA instruments, operating in uniaxial tensile mode. A 20 × 5 × 0.2 mm$^3$ film was vertically placed into the chamber, and the gauge length between the top and bottom grips was 15.41 mm. The runs were conducted with a controlled force protocol featuring a small applied uniaxial tensile force in the temperature range of 220–280 K, at a rate of −1 K min$^{-1}$. Isothermal loading and unloading were performed in the range of 0–10 N, at a rate of 1 N min$^{-1}$, within a temperature range 210–270 K, at 5 K temperature steps.

The thickness of the film was determined with a Sensofar PLμ 2300 optical imaging profiler.

## Data availability

The authors declare that the data supporting the findings of this study are available within the paper and its Supplementary Information files.

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

## Acknowledgements

Financial support from MCIN/AEI/10.13039/501100011033 (Spain) under Grant Nos. PCI2022-132957 (L.M., K.L.) (EU, MAT ERA.Net Program), PID2020-113549RB-I00/AEI (L.M.), PID2020-112975GB-I00 (J.L.T., P.L.), CEX2023-001300-M (J.L.T) and IJC2020-043957-I (E.S.), from AGAUR (Catalonia) under Projects Nos. 2021SGR00328 (L.M.) and 2021SGR00343 (J.L.T., P.L.) and from the Scientific and Technological Research Council of Turkey (TÜBITAK), Grant No: MFAG-121F151 (B.E.) are acknowledged.

## Author contributions

K.L. contributed to the magnetic, calorimetric and mechanical experiments; carried out the formal analysis of the data; and contributed to the writing, reviewing and editing of the manuscript. E.K., K.G., O.A. and B.E. were involved in sample preparation and characterisation. M.P. and A.P. were responsible for modelling. P.L. and J.L.T. carried out the calorimetry under pressure experiments. E.S.T. and E.K. were involved in the mechanical experiments. L.M. was responsible for the conceptualisation and writing of the original draft. B.E., E.S.T., A.P., J.L.T., P.L. and L.M. supervised the research and contributed to the review and editing of the manuscript. Funding acquisition corresponds to L.M., J.L.T, P.L. and B.E.

## Competing interests

The authors declare no competing interests.
