## [Peer Review File · Nature Communications]

Elastocaloric, Barocaloric and Magnetocaloric effects in Spin Crossover Polymer Composite FilmsREVIEWER COMMENTS

Reviewer #1 (Remarks to the Author):

The manuscript is well written, structured, and the conclusions are well argued. I think this article can be recommended for publication in a respected and highly rated journal. While the experimental design and presentation of results are commendable, there are areas where improvements could be made. I recommend minor revisions before accepting for publication.

Additional comments:

1. The abstract of the manuscript should be abbreviated and reflect the very essence of the study.
2. There is no formula/structure of the complex in the text. The absence of this information makes it difficult to understand the text.
3. There is no description of the molecular weight of the polymer used in the paper. There is a minimum of information on this object.
4. The process of interaction of the complex with the polymer is not quite clear. Is it a powder encapsulated in a polymer matrix? Or is the object of study the micelles of the complex stabilised by a polymer shell? I think the authors should think about this process and even illustrate it graphically.
5. The authors air-dried the films. So there is a chance that they are dealing with tetrahydrofuran solvates. Has that been clarified in any way?
6. In the course of reading the text, I did not find information on the percentage of polymer and complex in the resulting film. The technology of film production is written sparingly.
7. Why the film temperature is not essentially different from the phase transition temperature of the original complex? I would like to compare all three samples: complex-polymer-film. I would like to ask the authors not to cut out the figures and their captions from the main text of the manuscript. This makes it difficult to understand the text.

Recommendation: minor revision.

Reviewer #2 (Remarks to the Author):

Manosa and colleagues observed elastocaloric, barocaloric, and magnetocaloric effects in spin-crossover composite materials. They previously reported giant barocaloric effects in spin-crossover materials (Adv. Mater. 2019, 1807334; 2021, 2008076). However, the referee must say that the submission's preparation is poor, as it appears to be a working report attempting to contain too much data. More importantly, some key performances, such as the magnetocaloric effect, is not excellent. In addition, the caloric properties of the spin-crossover composites should be compared to those of PVC and $\text{Fe(L)}_2(\text{BF}_4)_2$. The author should provide multiple cycles of measurements to demonstrate the reversibility and reproducibility of the spin-crossover composites, while excluding the effect of sample preparation. Overall, this submission's status makes it difficult to meet the Nature Commun criteria. The referee strongly recommends that you reconstruct the story and resubmit it to a suitable journal.

Reviewer #3 (Remarks to the Author):

I read the manuscript with great interest. Klara and the co-authors demonstrate a more compact application of SCO materials in MCE, BCE, and eCE, via including the powder-like

SCO compounds into PVC matrix. The authors compared the performances of the 1/PVC and the pure SCO materials, in MCE, BCE, and eCE test, and reasonably modeled the different T_c and broadened transition range in eCE. I could see the novelty of the research, and there are few technique problems. The introduction part shows poor writing logic (at least to me), which fail in highlighting the significance and urgency of such research. Therefore, before the manuscript can get a recommendation for publication, the authors should perform major revisions on the manuscript. And I also list my other concernings as follows:

1. The included SCO particles occupy a ratio of 30%, the manuscript seems silent about why such ratio is selected. Have the authors tried other ratios?
2. I have some confusions about the modelling process in the discussion part. The model introduces disorders/randomness in B value for inclusions of different sizes, it seems that some quantified parameters are missing, for examples, mathematically, what types of randomness are introduced to fitting the broadening of the transition, a Gaussian distribution or other types.
3. Also about the modeling, it seems that T'_c is always lower than T_c according to $T'_c = T_c - 2B/A$, however, in Fig.5c, we note in 1/PVC sample, there is part in the broadened transition peak that features higher T'_c than T_c , why.
4. Baseline correction is used for obtaining the transition entropy change of 1/PVC, the details about the baseline's selection is missing.
5. Does the bulk modulus of PVC matrix possibly influence the eCE's intensity or the spin cross ratio of the inclusions? Please comment about this in the discussion, instructions on how we choose matrix materials for the compact SCO device for eCE should be included.
6. The large amount of residual HS states in 1/PVC even at temperature far below T_c is verified by X-ray, but without reasonable explanation on its mechanism, how the matrix hinders the spin cross when lowering the temperature below T_c .
7. Some minor revisions are also necessary, including typos in the abstract like 'effect'; B's definition is not right behind Equation (5) where it for the first time appears; English should be improved.

POINT-BY-POINT RESPONSE TO THE REFEREES

Reviewer#1

The manuscript is well written, structured, and the conclusions are well argued. I think this article can be recommended for publication in a respected and highly rated journal. While the experimental design and presentation of results are commendable, there are areas where improvements could be made. I recommend minor revisions before accepting for publication.

We appreciate very much the positive evaluation of the reviewer, and are happy that he/she considers the paper to deserve publication in a respected and highly rated journal. We address below the specific minor additional comments:

1. The abstract of the manuscript should be abbreviated and reflect the very essence of the study.

We have shortened the abstract and have highlighted the essence of the study.

2. There is no formula/structure of the complex in the text. The absence of this information makes it difficult to understand the text.

We have included additional information about the complex in the supplementary material (Section 1), including the formula of the complex.

3. There is no description of the molecular weight of the polymer used in the paper. There is a minimum of information on this object.

The polymer used in the study was PVC from Fluka Honeywell (product nr. 81392), CAS No. 900-86-2, with a molecular weight of 62.49822 g/mol, and a density of 1.4 g/ml (at 25°C).

These data have been included in the new version of the manuscript (Methods Section).

4. The process of interaction of the complex with the polymer is not quite clear. Is it a powder encapsulated in a polymer matrix? Or is the object of study the micelles of the complex stabilised by a polymer shell? I think the authors should think about this process and even illustrate it graphically.

The polymer is non-polar, the complex has an ionic structure, and the only interaction between the polymer and the complex is elastic in nature. The complex forms small particles which are dispersed within the polymer matrix (as shown by Scanning Electron Microscopy, Fig. S8).

5. The authors air-dried the films. So there is a chance that they are dealing with tetrahydrofuran solvates. Has that been clarified in any way?

The polymer was dissolved in THF during synthesis. It was then air-dried, and further dried in an oven at 50°C for a long time while mass loss was checked. Furthermore, the absence of THF was confirmed by thermogravimetry between ambient temperature and 120° C, in a nitrogen atmosphere (in Al pan).

We have added this information in the revised version of the manuscript (Methods section).

6. In the course of reading the text, I did not find information on the percentage of polymer and complex in the resulting film. The technology of film production is written sparingly.

The initial amount of SCO from $[\text{Fe}(\text{L})_2][\text{BF}_4]_2$ / PVC film was 30%. The percentage of SCO in the film was determined by Fe analysis in the complex. A piece of the sample was cut from the film, and it was digested in a mixture of HNO_3 and H_2O_2 . The amount of Fe was determined in the prepared solution by Atomic Absorption. The percentage of complex in the film was determined according to the molecular mass of the complex and the film mass. It was confirmed that the targeted 30% SCO value was achieved as a result of the film production (Methods Section).

In the new version of the manuscript we have provided more details on the film production.

7. Why the film temperature is not essentially different from the phase transition temperature of the original complex? I would like to compare all three samples: complex-polymer-film.

We apologise for not clearly understanding what the reviewer means by “film temperature”. The complex undergoes a phase transition at a certain temperature (the SCO transition temperature) but the film does not undergo any transition. To make this comparison clearer, we have measured the thermal and magnetic properties of pure PVC. Results are shown in the supplementary material (Section 4). They confirm the stability of the polymer within the studied temperature range. On the other hand, comparison between the pure complex and the SCO/PVC is shown all along the manuscript (and supplementary material).

I would like to ask the authors not to cut out the figures and their captions from the main text of the manuscript. This makes it difficult to understand the text.

We agree with the reviewer that it would be desirable to have the figures in the main text. Nevertheless, we have used the Nature Publishing templates which keep the figures cut out from the main text.

Reviewer#2

Manosa and colleagues observed elastocaloric, barocaloric, and magnetocaloric effects in spin-crossover composite materials. They previously reported giant barocaloric effects in spin-crossover materials (Adv. Mater. 2019, 1807334; 2021, 2008076). However, the referee must say that the submission's preparation is poor, as it appears to be a working report attempting to contain too much data.

The paper is the result of a thorough study of all possible caloric properties which can occur in SCO/polymer films. As such, the study results in a large amount of relevant information. We believe the reader will appreciate having a detailed and quantitative comparison of the different caloric effects. From that point of view we do not agree with the reviewer that the manuscript contain too much data.

More importantly, some key performances, such as the magnetocaloric effect, is not excellent.

The reviewer is right in stating that the performances of the magnetocaloric effect are limited. This was expected due to the weak magnetism of SCO compounds. However, the key point, strength and novelty of the paper refer to the mechanocaloric properties which are shown to be the best among reported caloric materials. As previously stated, the magnetocaloric effect is included for comparison.

A first relevant result is the demonstration that the unique barocaloric properties of SCO are maintained when the compound is dispersed within a polymeric matrix. This finding broadens the field of possible technological applications of the barocaloric effect in cooling technologies. It is worth remembering that giant and colossal barocaloric materials are powders, which makes their implementation in cooling devices challenging. SCO/PVC films are compact and flexible. Tailored shapes can be envisaged which will be more easily adapted and implemented into a cooling device.

An even more important and new result is that our work represents the first report of giant elastocaloric effects associated with a spin crossover transition. Importantly, it is shown that the potential elastocaloric performances of SCO/PVC are excellent. In particular, large entropy changes are obtained at very low values of applied stress, and for low values of strain. These characteristics provide SCO/polymer films a Coefficient of Performance one order of magnitude larger than any other elastocaloric material studied thus far. Notice that until now, elastocaloric materials were considered to display very poor Coefficients of Performance when compared to other caloric materials.

Finally, it is worth remarking that the ideas and techniques reported in our manuscript applied to spin crossover materials can be extensive to many other barocaloric materials, such as plastic crystals. Actually, it is worth noting that barocaloric materials display the largest entropy changes (approaching those of conventional fluid refrigerants) among any caloric material. However the design of barocaloric cooling devices is challenging and no prototypes are reported thus far, probably due to the difficulty of applying hydrostatic pressures (requirement of pressure transmitting medium, etc). Hence, the possibility of underpinning these colossal entropy changes by means of an easily applicable external stimulus such as uniaxial stress is very appealing. Our paper demonstrates that this possibility is feasible. We have shown that it is possible to take advantage of the unique colossal caloric properties of Spin Crossover Materials, but driving them by uniaxial stress instead of hydrostatic pressure. Indeed application of uniaxial stress is much simpler than hydrostatic pressure, and it is expected that highly performing cooling prototypes can be developed which use composites of barocaloric materials and polymers.

In addition, the caloric properties of the spin-crossover composites should be compared to those of PVC and Fe(L)2(BF4)2 .

We have undertaken a thorough study of the properties of PVC. In the new version of the manuscript (Supplementary Material, Section 4, Figures S4(a) and S4(b)) we have included calorimetric and magnetization data which show the good stability of the polymer within the studied temperature range.

In addition, we have also studied the elastocaloric properties of pure PVC. Results are also included in the new version of the Supplementary Material (Figure S4(c)). It is found that the elastocaloric effect of pure PVC is temperature independent, with entropy values which are one order or magnitude smaller than those found for SCO/PVC polymer.

The author should provide multiple cycles of measurements to demonstrate the reversibility and reproducibility of the spin-crossover composites, while excluding the effect of sample preparation.

We have studied the reversibility and reproducibility of the SCO/PVC composites. Results are included in the new version of the Supplementary Material (Section S5).

We have performed 10 calorimetric heating and cooling runs, and the obtained DSC curves fall exactly one on top of the other (see figure S5 (a)).

An excellent reproducibility is also observed from strain as a function of temperature measurements, as evidenced by Fig. S5(b), which shows results obtained for different values of applied load.

Reviewer#3

I read the manuscript with great interest. Klara and the co-authors demonstrate a more compact application of SCO materials in MCE, BCE, and eCE, via including the powder-like SCO compounds into PVC matrix. The authors compared the performances of the 1/PVC and the pure SCO materials, in MCE, BCE, and eCE test, and reasonably modeled the different T_c and broadened transition range in eCE. I could see the novelty of the research, and there are few technique problems.

We appreciate both, the positive evaluation of the reviewer and the fact that he/she acknowledges the novelty of the research. Below we provide a point-by-point answer to the technique problems and concerns.

The introduction part shows poor writing logic (at least to me), which fail in highlighting the significance and urgency of such research. Therefore, before the manuscript can get a recommendation for publication, the authors should perform major revisions on the manuscript.

We have re-written the introduction and we have tried to emphasize the significance and urgency of our work.

And I also list my other concernings as follows:

1. The included SCO particles occupy a ratio of 30%, the manuscript seems silent about why such ratio is selected. Have the authors tried other ratios?

The 30% ratio was the largest amount of SCO which we successfully managed to disperse into the polymer. This is now stated in the new version of the manuscript (Methods Section).

2. I have some confusions about the modelling process in the discussion part. The model introduces disorders/randomness in B value for inclusions of different sizes, it seems that some quantified parameters are missing, for examples, mathematically, what types of randomness are introduced to fitting the broadening of the transition, a Gaussian distribution or other types.

Randomness has been introduced by means of a Gaussian distribution of the values of the parameters T_c and B . This is now specified in the new version of the manuscript (Discussion Section).

3. Also about the modeling, it seems that T'_c is always lower than T_c according to $T'_c = T_c - 2B/A$, however, in Fig.5c, we note in 1/PVC sample, there is part in the broadened transition peak that features higher T'_c than T_c , why.

The reviewer is right in stating that T'_c is always lower than T_c . However, as there is some dispersion in the B values, the long tail of the Gaussian distribution allows the existence of negative B values, which lead to T'_c values larger than T_c . Strictly speaking, these values are unphysical (as B is related to the bulk modulus of the matrix). However, the model is only intended at reproducing qualitatively the decrease of the average value of T_c , and the broadening of the transition when the inclusions are embedded in the matrix. Actually, such an inconsistency could be corrected by using a distribution for B values which prevents the existence of negative values (e.g. a uniform distribution within a finite range). However, because that effect is very small, for simplicity we choose a Gaussian distribution.

We have added a comment along these lines in the revised version of the manuscript (Discussion Section).

4. Baseline correction is used for obtaining the transition entropy change of 1/PVC, the details about the baseline's selection is missing.

A convenient method to correct from the base-line in the integration of differential scanning calorimetry curves is presented in detail in J. Ortin, *Thermochim. Acta* 121 (1987) 397. In our measurements, a linear baseline (automatic correction provided by the TA Instruments Analysis Software) provides a very satisfactory correction. Below we include a figure showing an example of the thermal curve (black line) and the corresponding base-line correction (red line).

5. Does the bulk modulus of PVC matrix possibly influence the eCE's intensity or the spin cross ratio of the inclusions? Please comment about this in the discussion, instructions on how we choose matrix materials for the compact SCO device for eCE should be included.

The value of the polymer's bulk modulus does indeed influence the elastocaloric effect of the composite. For very stiff polymers (large bulk modulus) the elastic energy required to deform the matrix will be large. If the free energy difference between the HS and LS phase is not large enough to overcome the increase in (elastic) energy, the SCO transformation will be arrested and no elastocaloric effect will be obtained. On the other hand, very soft polymers (low bulk modulus) will be unable to transmit the external applied force to the SCO particles, and no elastocaloric effect associated with the SCO transition will occur. Hence, the bulk modulus of the matrix has to be within an appropriate range of values such that the polymer is capable of transferring the external force but also does not impede the transformation of the SCO particles.

We have added a paragraph along these lines in the revised version of the manuscript (Discussion Section).

6. The large amount of residual HS states in 1/PVC even at temperature far below T_c is verified by X-ray, but without reasonable explanation on its mechanism, how the matrix hinders the spin cross when lowering the temperature below T_c .

The occurrence of residual HS states at low temperatures far below T_c is quite common in SCO complexes embedded in elastic matrices, and it is mostly attributed to surface

effects (Raza et al. Chem. Comm. 47 (2011) 11501; Martinez et al. Chem. Mater. 22 (2010) 4271.)

We have included a sentence along these lines in the new version of the manuscript (Results Section).

7. Some minor revisions are also necessary, including typos in the abstract like 'effect'; B's definition is not right behind Equation (5) where it for the first time appears;

We thank the reviewer for calling our attention to these mistakes which have already been fixed in the new version of the manuscript.

English should be improved.

The revised version of the manuscript has been language edited by the Nature Research Editing Service.

REVIEWERS' COMMENTS

Reviewer #1 (Remarks to the Author):

The authors have conducted substantial and notable research on the manuscript text. In its current form, this article is suitable for publication in the Journal. I extend my sincere apologies to the authors for the misunderstanding regarding my intention to compare the temperature data of the complex itself, the polymer, and the film produced by the polymer containing the complex. It is important to note, however, that the proposed variant by the authors at this juncture offers a comprehensive clarification of the situation.

Reviewer #2 (Remarks to the Author):

The authors have responded to my questions and thoroughly revised the paper. I suggest that the editor consider accepting the paper.

Reviewer #3 (Remarks to the Author):

my concerns have been well addressed by the authors, the manuscript have been revised accordingly.

I recommend the publication of the manuscript